# Inflammatory osteolysis is regulated by site-specific ISGylation of the scaffold protein NEMO

Naga Suresh Adapala[1†], Gaurav Swarnkar[1†], Manoj Arra[1†], Jie Shen[1], Gabriel Mbalaviele[2], Ke Ke[1], Yousef Abu-Amer[1,3]*

[1]Department of Orthopaedic Surgery and Cell Biology & Physiology, Washington University School of Medicine, St. Louis, United States; [2]Bone and Mineral Division, Department of Medicine, Washington University School of Medicine, St. Louis, United States; [3]Shriners Hospital for Children, St. Louis, United States

**Abstract** Inflammatory osteolysis is governed by exacerbated osteoclastogenesis. Ample evidence points to central role of NF-κB in such pathologic responses, yet the precise mechanisms underpinning specificity of these responses remain unclear. We propose that motifs of the scaffold protein IKKγ/NEMO partly facilitate such functions. As proof-of-principle, we used site-specific mutagenesis to examine the role of NEMO in mediating RANKL-induced signaling in mouse bone marrow macrophages, known as osteoclast precursors. We identified lysine (K)270 as a target regulating RANKL signaling as K270A substitution results in exuberant osteoclastogenesis in vitro and murine inflammatory osteolysis in vivo. Mechanistically, we discovered that K270A mutation disrupts autophagy, stabilizes NEMO, and elevates inflammatory burden. Specifically, K270A directly or indirectly hinders binding of NEMO to ISG15, a ubiquitin-like protein, which we show targets the modified proteins to autophagy-mediated lysosomal degradation. Taken together, our findings suggest that NEMO serves as a toolkit to fine-tune specific signals in physiologic and pathologic conditions.

**\*For correspondence:**
abuamery@wustl.edu

[†]These authors contributed equally to this work

**Competing interests:** The authors declare that no competing interests exist.

## Introduction

The transcription factor NF-κB is expressed ubiquitously in all cell types, readily activated by numerous factors and cytokines (*Abu-Amer and Faccio, 2006*; *Hayden, 2004*; *Ravid and Hochstrasser, 2004*; *Ting and Endy, 2002*);it plays critical roles in modulating inflammation, immunity, cell proliferation, differentiation, and survival. While baseline NF-κB activity is essential for physiologic functions such as skeletal development (*Abu-Amer, 2013*; *Courtois et al., 2001*; *Häcker and Karin, 2006*; *Li et al., 2002*; *Ruocco and Karin, 2005*; *Ruocco and Karin, 2007*), exacerbated and chronic unrestrained activity of this transcriptional factor during inflammation leads to undesired harmful effects with major dysfunctional consequences including osteolysis (*Abu-Amer, 2013*; *Boyce et al., 2010*; *Pasparakis, 2008*; *Ruocco and Karin, 2005*; *Schett and David, 2010*; *Xing et al., 2005*; *Xu et al., 2009*). In this regard, we and others have shown that, whereas baseline activity of the principal NF-κB kinase IKKβ (also known as IKK2) is essential for normal skeletal development, its hyper and prolonged activation is pathologic (*Otero et al., 2010*; *Zhang et al., 2013*). In addition, NF-κB is the primary pathway that mediates inflammatory responses of numerous bone-targeting cytokines such as TNFα, IL-1β, and IL6 (*Abu-Amer, 2009*).

When a stimulus is provided, signaling molecules are recruited to distal domains of the appropriate receptor leading to the assembly of the IKK complex which includes IKKγ/NEMO and IKKβ, among other adaptor proteins. This process leads to phosphorylation of downstream substrates, most notably, IκBα and activation of downstream transcriptional machinery (*Boyce et al., 2005*;

**eLife digest** The human skeleton contains over 200 bones that together act as an internal framework for the body. Over our lifetime, the body constantly removes older bone tissue from the skeleton and replaces it with new bone tissue. This "bone remodeling" also controls how bones are repaired after being damaged by injuries, disease or normal wear and tear.

Cells known as osteoclasts are responsible for breaking down old bone tissue and participate in repairing damaged bone. A cellular pathway known as NF-kB signaling stimulates other cells called "bone marrow macrophages" to become osteoclasts. A certain level of NF-kB signaling is required to maintain a healthy skeleton. However, under certain inflammatory conditions, the level of NF-kB signaling becomes too high causing hyperactive osteoclasts to accumulate and inflict severe bone breakdown. This abnormal osteoclast activity leads to eroded and fragile bones and joints, as is the case in diseases such as rheumatoid arthritis and osteoporosis.

Previous studies have shown that a protein called NEMO is a core component of the NF-kB signal pathway, but the precise role of NEMO in the diseased response remained unclear. Adapala, Swarnkar, Arra et al. have now used site-directed mutagenesis approach to study the role of NEMO in bone marrow macrophages in mice. The experiments showed that one specific site within the NEMO protein, referred to as lysine 270, is crucial for its role in controlling osteoclasts and the breakdown of bone tissue. Mutating NEMO at lysine 270 led to uncontrolled NF-kB signaling in the bone marrow macrophages. Further experiments showed that lysine 270 served as a sensor to allow NEMO to bind another protein called ISG15, which in turn helped to decrease NF-kB signaling and slow down the erosion of the bone.

These findings suggest that site-specific targeting of NEMO, rather than inhibiting the whole NF-kB pathway, may help to reduce the symptoms of bone disease while maintaining the beneficial roles of this essential pathway. However, additional research is required to identify NEMO sites responsible for controlling the inflammatory component.

Boyce et al., 2010; Franzoso et al., 1997). Gene deletion studies have shown that members of the NF-κB signal transduction pathway, which include NF-κB1 (p50), NF-κB2 (P52), RelA (p65), IκBα, IKKα, IKKβ, and NEMO, are crucial for normal development of osteoclasts and their survival, and are considered as the principal mediators of RANK signaling (Boyce et al., 1999; Boyce et al., 2010; Franzoso et al., 1997). Despite the intense research efforts focusing on the role of this pathway in physiologic and inflammatory responses, little is known regarding the cell-specific response to a given signal and pairing it with corresponding function in homeostatic and pathologic settings. For example, whereas RANKL, TNFα, IL-1β, and other factors activate NF-κB in osteoclast precursors, the molecular machinery that assigns unique signaling signatures for each stimulus remains vague.

The NF-κB pathway offers a toolbox that enables signal and cell-specific signaling cascades. In this regard, signal specificity can occur at proximal (juxtaposed to receptors) and distal (downstream) pathway sites. In the former case, ligation of cell surface receptors such as TNFα receptor (TNFR), RANK or IL1 receptor (IL-1R) triggers a chain of events that leads to the formation of a signalsome that includes unique TRAF proteins, IKKα/β and NEMO. It has been further shown that this complex is regulated by post-translational machineries, chiefly phosphorylation, ubiquitination and SUMOylation (Chen, 2012; Ikeda et al., 2010; Ikeda et al., 2011; Laplantine et al., 2009; Liu and Chen, 2011; Ni et al., 2008; Sebban et al., 2006; Shambharkar et al., 2007; Yeh, 2009), which are believed to contribute significantly to signal specificity by directing downstream cues.

NEMO, a key player of the IKK signalsome, is a scaffold protein that lacks enzymatic activity; yet, it is essential for NF-κB signaling evident by convergence of upstream signals directed by TRAFs prior to assembly of downstream IKK signals (Li et al., 2001; May et al., 2002; Prajapati and Gaynor, 2002; Yamamoto et al., 2001). Recent studies have shown that specific NEMO domains and numerous lysine residues throughout the different domains of NEMO, especially the ubiquitin and zinc finger domains, undergo extensive ubiquitination, SUMOylation, and other post-translational modifications (PTMs) in response to various stimuli (Cordier et al., 2009; Hay, 2004; May et al., 2002; Rushe et al., 2008; Schröfelbauer et al., 2012; Sebban et al., 2006; Wu et al., 2006). Specifically, these domains and lysine residues serve as specific docking sites utilizing PTM moieties to

enable recruitment of unique signaling complexes and pathway substrates in one hand, and protea-some-mediated degradation, in the other hand. Indeed, the critical role of a number of lysines and other residues such as K270, K302, K312, K392, C417 in cellular signaling have been described (*Alhawagri et al., 2012*; *Bloor et al., 2008*; *Ni et al., 2008*; *Yang et al., 2004*). In this regard, series of NEMO mutants at specific lysine residues located at the coil zipper domain revealed dominant negative and constitutive activation properties of NEMO (*Bloor et al., 2008*).

In the current study, we tested the functional significance of key lysine residues individually in the ubiquitin and zinc finger domains of NEMO in response to RANKL. This approach was designed to test our hypothesis that certain NEMO lysine residues serve as signal-specific docking sites that facilitate the assembly of unique signal activating- or suppressing-protein complexes in cell and stimulus specific manner.

## Results

### NEMO$^{K270A}$ mutant expression in bone marrow macrophages exacerbates RANKL-induced osteoclastogenesis

To address our aforementioned hypothesis, we conducted broad lysine (K) screen of NEMO and substituted strategic K and D residues in tandem with alanines and asparagine, as indicated, (*Figure 1A*), to disrupt post-translational modifications of specific NEMO lysines, and hence, impede assembly of protein complexes and alter subsequent signaling. Wild type (WT) NEMO (NEMO$^{WT}$) and various NEMO K mutants (NEMO$^{K}$) were cloned in pMx-retroviral plasmid and expression of representative clones was confirmed in HEK293 cells (*Figure 1B*). Viral particles of these plasmids were produced using PLAT-E cells as described previously (*Swarnkar et al., 2016*). Protein expression of NEMO$^{WT}$ and NEMO$^{K}$ mutants was conducted in primary bone marrow macrophages (BMMs a.k.a osteoclast precursors). BMMs from WT or NEMO null cells (NM-cKO) expressing NEMO$^{K270A}$ formed ample osteoclasts (OC), reminiscent of inflammatory condition, upon exposure to permissive concentration of RANKL, when compared with cell expressing NEMO$^{WT}$ (NM-WT) (*Figure 1C*) or other forms of NEMO mutants (*Figure 1—figure supplement 1A*), suggesting sensitization of RANKL signaling in the absence of K270 residue in NEMO. These observations were further supported by increased OC surface area (*Figure 1D*), elevated expression of osteoclast differentiation markers (*Figure 1E–J*) and NF-κB activity (*Figure 1K*) in RANKL-treated NEMO$^{K270A}$ expressing BMMs compared with NEMO$^{WT}$ (NM-WT) and other NEMO forms. Expression of NEMO$^{K270A}$ (also referred to in figure labels as NM-KA for brevity) in BMMs was robust and stable compared with NM-WT and other NEMO mutants (*Figure 1—figure supplement 1B*), despite infection of BMMs with equal number of viral particles. In fact, hyper osteoclastogenesis by NEMO$^{K270A}$ was not due to higher expression of the transgenic NEMO protein because 1:20 dilution of viral particles which gave rise to protein expression levels approximating those of NEMO$^{WT}$ and of other NEMO constructs still provoked heightened osteoclastogenesis (*Figure 1—figure supplement 1A–B*). These findings support the notion that intact K270 appears to be essential to restrain RANKL-induced excessive osteoclastogenesis.

### Expression of NEMO$^{K270A}$ in vivo leads to inflammatory osteolysis and joint pathology

To examine the functional relevance of this in vitro abnormal signaling by NEMO$^{K270A}$ mutant in the in vivo state, we generated NEMO$^{K270A}$ transgenic mice by cloning this construct in the *Gt(ROSA) 26Sor* locus as described previously (*Figure 2—figure supplement 1A*; *Otero et al., 2012*; *Swarnkar et al., 2014*). *Gt(ROSA)26Sor* harboring NEMO$^{WT}$ (*ikbkg*) was also generated as a control (*Figure 2—figure supplement 1B*). Mice were born at mendelian ratio, yet transgenic NEMO$^{K270A}$ (*ikbkg*) knockin (KA) mice were significantly smaller in size compared with transgenic WT and control mice (*Figure 2A*). Upon closer analysis, we observed significant joint swelling (*Figure 2A*; arrows), splenomegaly, altered hematopoiesis (*Figure 2B* and *Figure 3—figure supplement 1*), and enhanced mortality at 6–8 weeks of age (not shown). Micro-computed tomography (uCT) scans and X-ray images revealed dramatic bone loss reaching 50% of BV/TV in severe cases (*Figure 2C–I* and *Figure 2—figure supplement 1C–F*), with no such loss in NEMO$^{WT}$ transgenic mice (NM-WT-Tg) (*Figure 2—figure supplement 1G–L*). Notably, knee and ankle joints were severely damaged

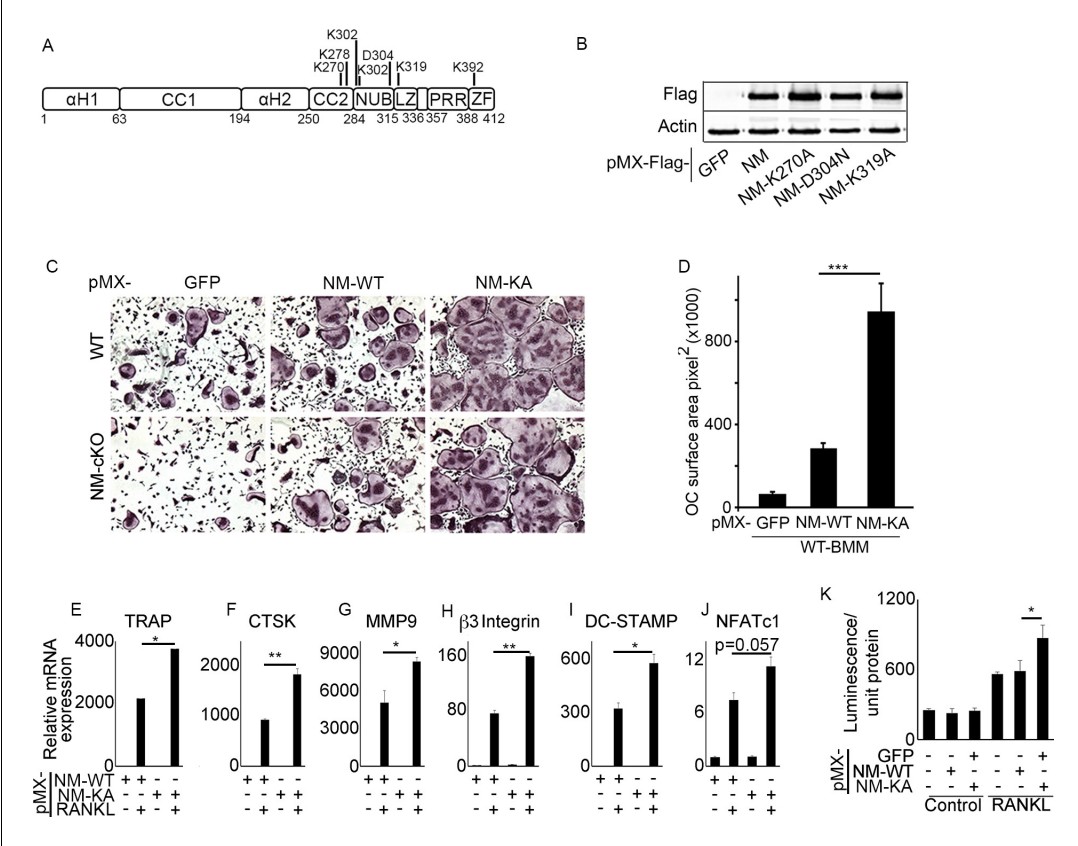

**Figure 1.** NEMOK270A mutant expression in BMMs exacerbates RANKL-induced osteoclastogenesis. (**A**) Domain structure of NEMO (**B**) Western blot showing expression of *pMX-NEMO$^{WT}$* (NM) and *pMX-NEMO* mutants (NM-K270A, NEMO-D304N and NM-K319A). (**C**) BMMs from WT and (*LysM-cre-NEMO f/f*) NEMO-cKO mice were transduced with viral particles (generated by transfecting pMX- retroviral vectors in PLAT-E cells) expressing NEMO$^{WT}$ (NM-WT) and NEMO$^{K270A}$ (NM-KA) and cultured in the presence of MCSF (10 ng/ml) and RANKL (50 ng/ml). (**D**) Representative TRAP staining for osteoclast (n = 8) and (**D**) quantification of TRAP positive OCs. qPCR analysis for OC marker genes (**E**) *TRAP*, (**F**) *CTSK*, (**G**) *MMP9*, (**H**) *β3integrin*, (**I**) *DC-STAMP* and (**J**) *NFATC1 (p=0.057)*. Representative data (n = 3 independent experiments). (**K**) BMMs from RelA_luc reporter mice expressing NM-WT and NM-KA were cultured in the presence of MCSF (10 ng/ml) for 3 days followed by RANKL stimulation with RANKL (50 ng/ml) for 6 hr and RelA-luciferase activity measurement (n = 3). *pMX-Flag-NEMO$^{WT}$-RFP* (NM-WT), *pMX-Flag-NEMO$^{K270A}$-RFP* (NM-KA). (*p<0.05, **p<0.01 and ***p<0.001). The online version of this article includes the following source data and figure supplement(s) for figure 1:

**Source data 1.** Western blot showing expression of *pMX-NEMOWT*(NM) and*pMX-NEMO*mutants (NM-K270A, NEMO-D304N and NM-K319A).
**Source data 2.** qPCR analysis for OC marker genes.
**Source data 3.** RelA-luciferase activity.
**Figure supplement 1.** BMMs from wild type mice were transduced with viral particles (generated by transfecting pMX- retroviral vectors in PLAT-E cells) expressing NEMO$^{WT}$ (NM-WT), NEMO$^{K270A}$ (NM-KA), NEMO-D304N and NEMO-K319A constructs followed by culture in the presence of MCSF (10 ng/ml) and RANKL (50 ng/ml) for 4 days.

(*Figure 2D*:compare WT with KA, *Figure 2—figure supplement 1C,E,F*:arrows). Histological analysis showed that bone and joint sections contain exuberant number of TRAP-positive OCs with evidence of loss of trabecular bone and destruction of articular cartilage by osteoclasts and inflammatory cells, which may also result from synovial pannus formation containing osteoclasts (*Figure 2J–K*: arrows). This phenotype was further corroborated with increased circulating serum levels of TRAcP5b and carboxyl-terminal cross-linked telopeptide of type one collagen (CTX-1), both are well established markers of bone breakdown (*Figure 2L–M*).

## NEMO$^{K270A}$ mutation instigates systemic inflammation

The observed joint swelling, skeletal degeneration, splenomegaly, and osteolysis point to systemic inflammation and altered hematopoiesis. Indeed, multiplex ELISA revealed that NEMO$^{K270A}$ mice express copious amounts of circulating inflammatory cytokines and chemokines in the serum

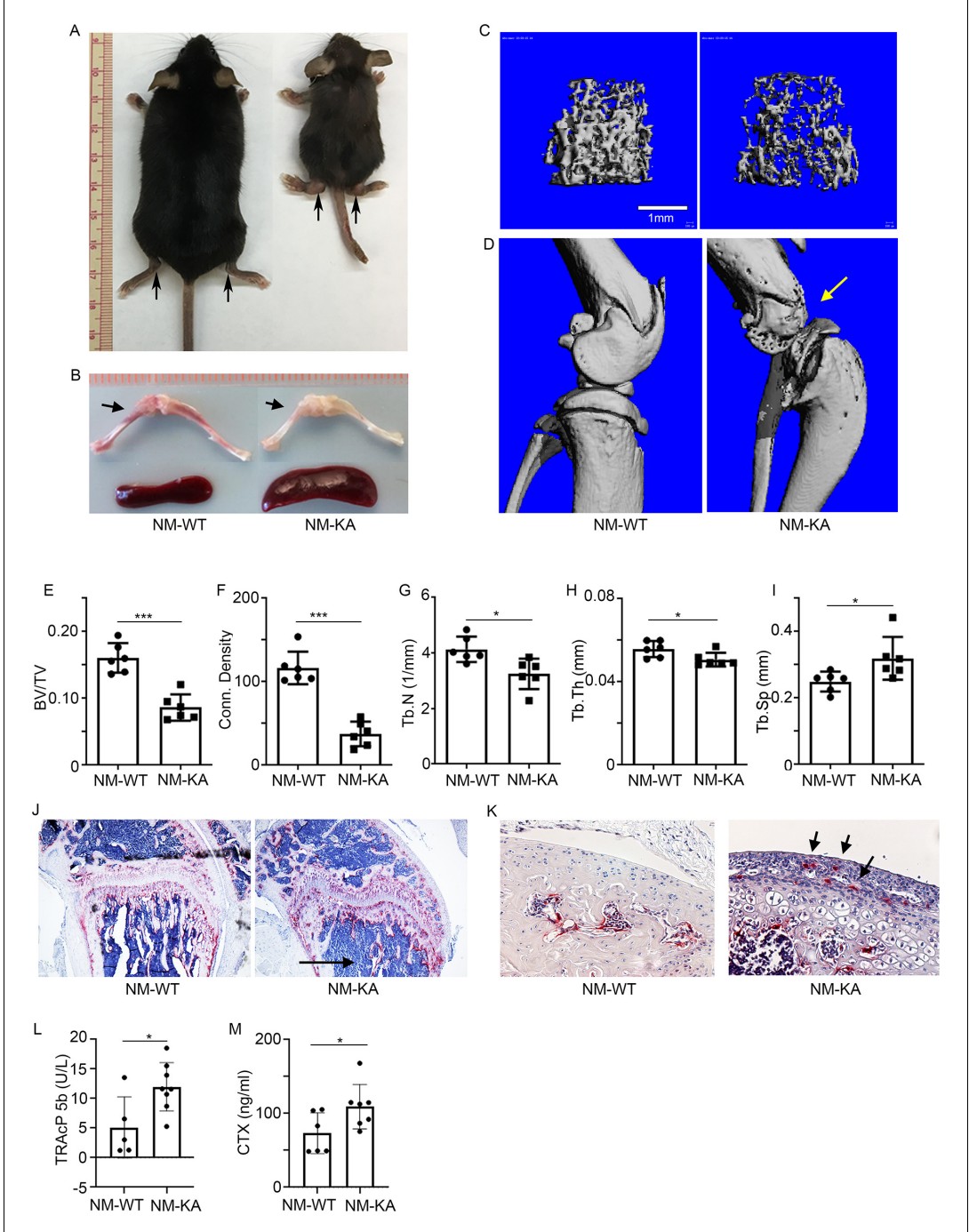

**Figure 2.** Expression of NEMO$^{K270A}$ in vivo leads to inflammatory osteolysis and joint destruction. *NEMO$^{K270A}$* was conditionally expressed in myeloid cells (NM-KA mice) by crossing *NEMO$^{K270A}$ f/f* mice with *LysozymeM cre* expressing mice. (**A**) Whole body images of NM-KA mice compared to littermate wild type control mice (6 weeks old). The arrows point to deformed joints and swelling. (**B**) Photomicrograph of spleen and bone from NM-WT and NM-KA mice. MicroCT analysis of bone from NM-WT and NM-KA mice showing (**C**) femur trabecular bone, (**D**) knee joint osteolysis (arrow) and quantification of (**E**) Bone volume/total volume (BV/TV), (**F**) Connectivity density, (**G**) Trabecular number (Tb.N), (**H**) Trabecular thickness (Tb.Th) and (**I**) Trabecular separation (Tb.Sp) in the femur trabecular region (n = 6). Long bones from 6 weeks old NM-WT and NM-KA mice were processed for histology and stained for TRAP to visualize TRAP+ osteoclasts in (**K**) bone sections and (**K**) Articular surfaces of knee joint (arrow). Representative images (n = 6) Serum was collected from NM-WT and NM-KA to measure serum (**L**) TRAP and (**M**) CTX concentration as an indicator of increased osteoclast activity (n = 6–8). *LysM-cre-NEMO$^{WT}$-f/f* (NM-WT), *LysM-cre-NEMO$^{K270A}$-f/f* (NM-KA) mice. (*p<0.05, **p<0.01 and ***p<0.001).
The online version of this article includes the following source data and figure supplement(s) for figure 2:

**Source data 1.** MicroCT analysis of bone from NM-WT and NM-KA mice.
*Figure 2 continued on next page*

*Figure 2 continued*

**Source data 2.** Serum concentration of TRAP and CTX.
**Figure supplement 1.** Generation of NEMO transgenic mice.

(*Figure 3A–L*. At the cellular level, FACS analysis confirmed skewing of hematopoiesis toward myelopoiesis evident by abundant frequency of CMPs and GMPs, the immediate OC progenitors, in spleen and bone marrow compartments (*Figure 3—figure supplement 1A–E*; arrows). In addition, we detected a spike in neutrophils frequency in NEMO$^{K270A}$ knockin mice (*Figure 3—figure supplement 1F–J*; arrows). These observations suggest that the ensuing inflammatory microenvironment in these mice alters hematopoiesis and exacerbates osteoclastogenesis. To provide additional mechanistic support for this proposition, ex-vivo osteoclastogenesis experiments showed that BMMs derived from NEMO$^{K270A}$ knockin mice readily formed osteoclasts under RANKL permissive conditions (*Figure 3M*). Furthermore, expression of osteoclastogenic markers (*Figure 3N–R*) and activation of NF-κB (p-p65/RelA) (*Figure 3S*) were markedly elevated in RANKL-stimulated NEMO$^{K270A}$ compared with WT cells.

## NEMO$^{K270A}$ mutation hampers autophagy in BMMs

To glean more mechanistic insights, we examined localization and cellular distribution of NEMO in HEK293 (PLAT-E) cells and BMMs. Whereas NEMO$^{WT}$ was evenly diffused in the cytoplasm, NEMO$^{K270A}$ transgene formed puncta juxtaposed to nuclei (*Figure 4A*; arrows) in PLAT-E cells. Electron microscopy scanning of RANKL-treated BMMs (x7,500 magnification) further showed that unlike WT cells, NEMO$^{K270A}$ cells (panel labeled NM-KA) exhibit cytoplasmic aggregates (*Figure 4—figure supplement 1A*; yellow arrows), reminiscent of cytoplasmic debris. Thus, we surmised that K270A mutation of NEMO may have altered physiologic autophagy. To this end, in vitro RANKL-primed pre-OCs expressing NEMO$^{WT}$ and NEMO$^{K270A}$ fixed and stained with mouse NEMO and rabbit LC3 antibodies and Alexa Fluor secondary conjugates. LC3 is a *bona-fide* marker of autophagy, which is lapidated and degraded during physiologic autophagy. In contrast, sustained elevated levels of LC3 are observed during dysfunctional autophagy. The data summarized in *Figure 4B* and *Figure 4—figure supplement 1B,C*, showed that NEMO$^{K270A}$ and LC3 accumulation was far greater than NEMO$^{WT}$ and LC3 (arrows). This result was further mirrored by LC3 immunoblots (*Figure 4C*), pointing to a possible defect in autophagy flux in NEMO$^{K270A}$ expressing cells. Even more convincingly, quantitative flow cytometry analysis of LC3-GFP levels in NEMO$^{WT}$ and NEMO$^{K270A}$ BMMs revealed marked accumulation of this protein in the latter cells (black dots) compared with WT cells (black dots) 6 hr post starvation (*Figure 4D*). Further analysis depicted in *Figure 4E* showed that starvation (yellow histograms) led to accumulation of LC3 in NEMO$^{K270A}$ cells (shifted to the right) but not in WT cells (shifted to the left), which was reversed in WT cells in the presence of autophagy inhibitor chloroquine (pink histograms) to mimic NEMO$^{K270A}$ cells, as expected. In fact, chloroquine exacerbated osteoclastogenesis by WT just as NEMO$^{K270A}$ did (*Figure 4—figure supplement 1D*). Finally, a change in mean fluorescence intensity (MFI) of GFP showed pronounced decrease in LC3 in NEMO$^{WT}$ (NM-WT) compared with meager change in NEMO$^{K270A}$ (NM-KA) cells (*Figure 4F*). Consistent with these observation, levels of mTOR, a well-documented autophagy regulator, were elevated in NEMO$^{K270A}$ (NM-KA) cells (*Figure 4—figure supplement 1E*). Taken together, quantitative FACS scatters and histograms confirm that NEMO$^{K270A}$ cells have a defect in autophagy flux and aggregation of mutant NEMO that lends itself to heightened signaling.

Accumulation of NEMO$^{K270A}$ aggregates (puncta) suggests a potential defect in lysosomal degradation due to restriction in autophagosomes. Thus, we examined localization of NEMO with the lysosome marker LAMP1 by immunofluorescence and by EM scanning. The data show that NEMO$^{K270A}$ failed to localize with lysosomes (*Figure 5A–B*, *Figure 5—figure supplement 1A–C*), suggesting a defect in delivery of NEMO$^{K270A}$ in autophagosome to lysosome. In fact, careful examination in EM images, showed robust expression of NEMO$^{K270A}$ which was restricted to autophagosome (AP) structures compared to NEMO$^{WT}$ observed in lysosomes (L) (*Figure 5B*). Finaly, we show that following RANKL stimulation of BMMs and subsequent starvation, NEMO puncta accumulation persisted in NEMO$^{K270A}$ cells following serum starvation (akin to defective autophagy flux) yet puncta faded in WT cells following starvation as it undergoes processing by normal autophagy flux (*Figure 5C–D*). In

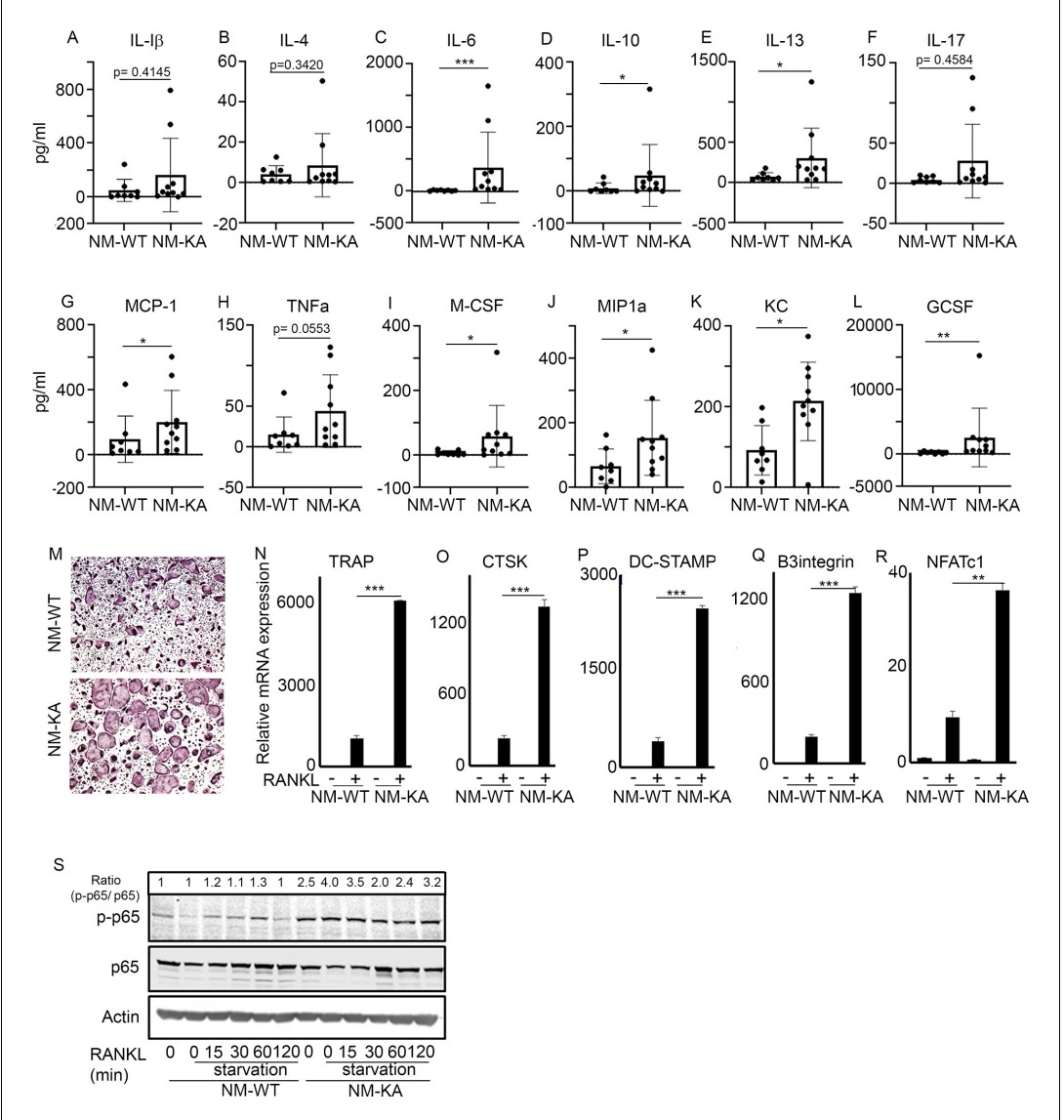

**Figure 3.** NEMO$^{K270A}$ mutation instigates systemic inflammation. Serum was collected from NM-WT and NM-KA mice (n = 8–10) to measure concentration of inflammatory cytokines (A) Interleukin (IL) 1b, (B) IL-4, (C) IL-6, (D) IL-10, (E) IL-13, (F) IL-17, (G) Monocyte chemoattractant protein1 or CCL2, (H) Tumor necrosis factor alpha, (I) Macrophage colony stimulating factor, (J) macrophage Inflammatory protein (MCP)−1 or CCL3, (K) keratinocyte chemoattractant or neutrophil activating protein three or CXCL1 and (L) granulocyte colony stimulating factor (GCSF). (M) BMMs from NM-WT and NM-KA mice were isolated and cultured in the presence of MCSF (10 ng/ml) and RANKL (10 ng/ml). Representative TRAP staining for osteoclast (n = 8) is shown. (N–R) Representative qPCR analysis for OC marker genes *TRAP, CTSK, β3integrin, DC-STAMP* and *NFATC1* (n = 3). (S) BMMs from NM-WT and NM-KA mice were isolated and cultured in the presence of MCSF (10 ng/ml) four days followed by serum starvation and stimulation with RANKL (50 ng/ml) for different time points (n = 8). Representative western-blot showing activation of p65 (phos-p65/p65 ratio) post RANKL stimulation in BMMs from NM-WT and NM-KA mice. *LysM-cre-NEMO-WT-f/f* (NM-WT), *LysM-cre-NEMO-K270A-f/f* (NM-KA) mice. (*p<0.05, **p<0.01 and ***p<0.001).

The online version of this article includes the following source data and figure supplement(s) for figure 3:

**Source data 1.** Serum concentration of cytokines from NM-WT and NM-KA mice measured by ELISA.
**Source data 2.** Representative qPCR analysis for OC marker genes.
**Source data 3.** Representative western-blot of p65 (phos-p65/p65 ratio) post RANKL stimulation in BMMs from NM-WT and NM-KA mice.
**Figure supplement 1.** BrdU was injected to NM-WT and NM-KA mice.

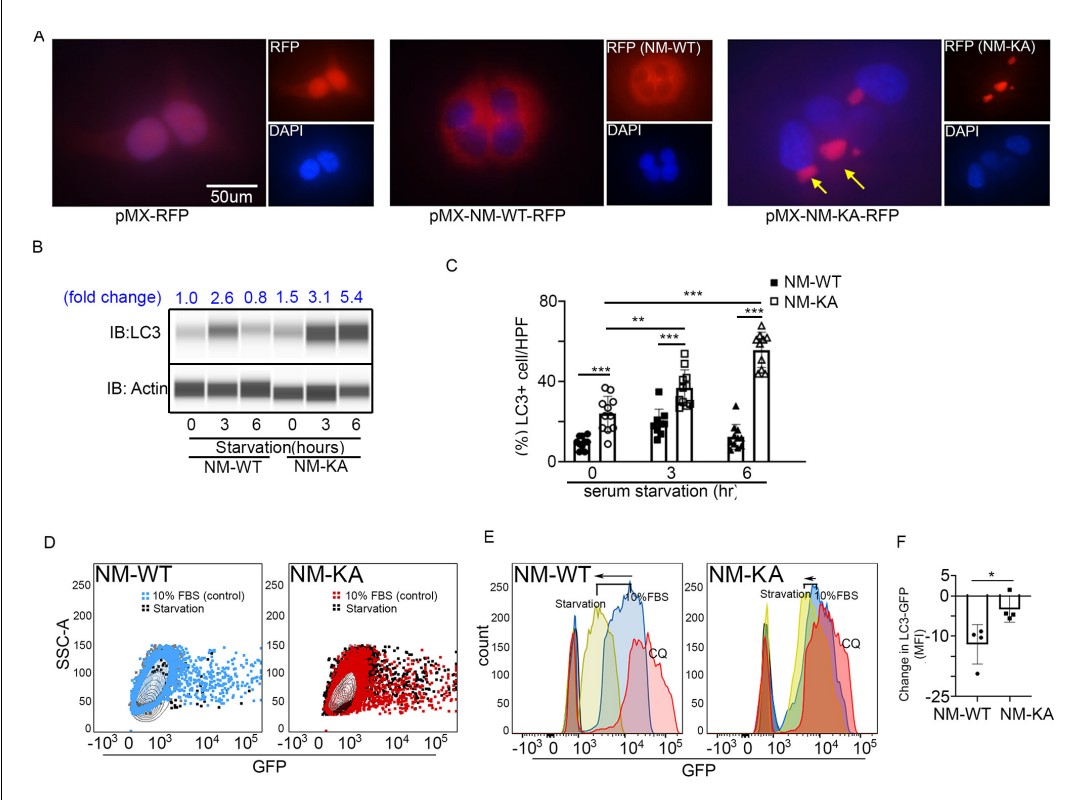

**Figure 4.** NEMO[K270A] mutation hampers autophagy. PLAT-E cells were transfected with retroviral *pMX-Flag-NEMO-[WT]-RFP* (NM-WT) and *pMX-flag-NEMO-[K270]-RFP* (NM-KA) expression vector. (**A**) Fluorescence images showing distribution of NM-WT-RFP in cytoplasm compared to puncta (yellow arrows) (juxtaposed to nuclei- DAPI stained) formation in case of NM-KA-RFP in PLAT-E cells. (**B**) Western blot for LC3 using WES (protein simple). BMMs were cultured for 2 days with RANKL (preOC) followed by 6 hr of serum starvation and western blotting. Fold change of LC3 relative to actin is indicated on top. (**C**) Quantification of LC3+ cells per high magnification field. (**D**) For flow cytometry, BMMs were transduced with pMX-GFP-LC3-RFP retrovirus generated in PLAT-E packing cells, and flow analysis was done to detect GFP signal or LC3 flux. Contour plots showing LC3-GFP+ expressing cells in NM-WT and NM-KA preOC (Blue: NM-WT without serum starvation, Red: NM-KA without serum starvation, and Black: after 6 hr of serum starvation), (**E**) Histograms representing shift in LC3-GFP+ cells following induction of autophagy (Red histogram: background signal in uninfected cells, Blue histogram: No serum starvation or 10% FBS control, yellow: 6 hr serum starvation, and pink: chloroquine), (**F**) Change in Mean fluorescent intensity (MFI) showing LC3-GFP signal in NM-WT and NM-KA preOC cells post autophagy induction. *LysM-cre-NEMO-WT-f/f* (NM-WT), *LysM-cre-NEMO-K270A-f/f* (NM-KA) mice. (*p<0.05). (*p<0.05, **p<0.01 and ***p<0.001).

The online version of this article includes the following source data and figure supplement(s) for figure 4:

**Source data 1.** Western blot for LC3 using WES (protein simple).
**Source data 2.** Quantification of LC3+ cells.
**Source data 3.** LC3-GFP FACS analysis.
**Figure supplement 1.** Autophagy is negatively impacted in NEMO[K270A] cells.

sum, our data suggest that NEMO mutation at K270A disrupts the delivery of NEMO[K270A] from autophagosome to lysosomes leading to accumulation of NEMO signals and subsequent buildup of inflammatory and osteoclastogenic signals.

*Intact NEMO K270 residue is essential for post-translational modification (PTM) to regulate autophagy and osteoclastogenesis.* To further elucidate the mechanisms underpinning inflammation and osteolysis in NEMO[K270A] transgenic mice, we conducted proteomics on NEMO[WT] and NEMO[K270A] immunoprecipitates. Proteomic analysis revealed that expression of autophagy and PTM proteins, especially the ubiquitin-like protein ISG15, is altered in cells expressing NEMO[K270A] mutant compared with WT cells (*Figure 6A*). Note that ISG15 is an IFN-stimulated gene and a ubiquitin-like protein that modulate cellular signals through a process termed ISGylation analogous to ubiquitination, but its function during osteoclastogenesis has not been described. Indeed, immunofluorescence images displayed NEMO(red)-ISG15(green) co-localization in vacuolar structures in NEMO[WT]

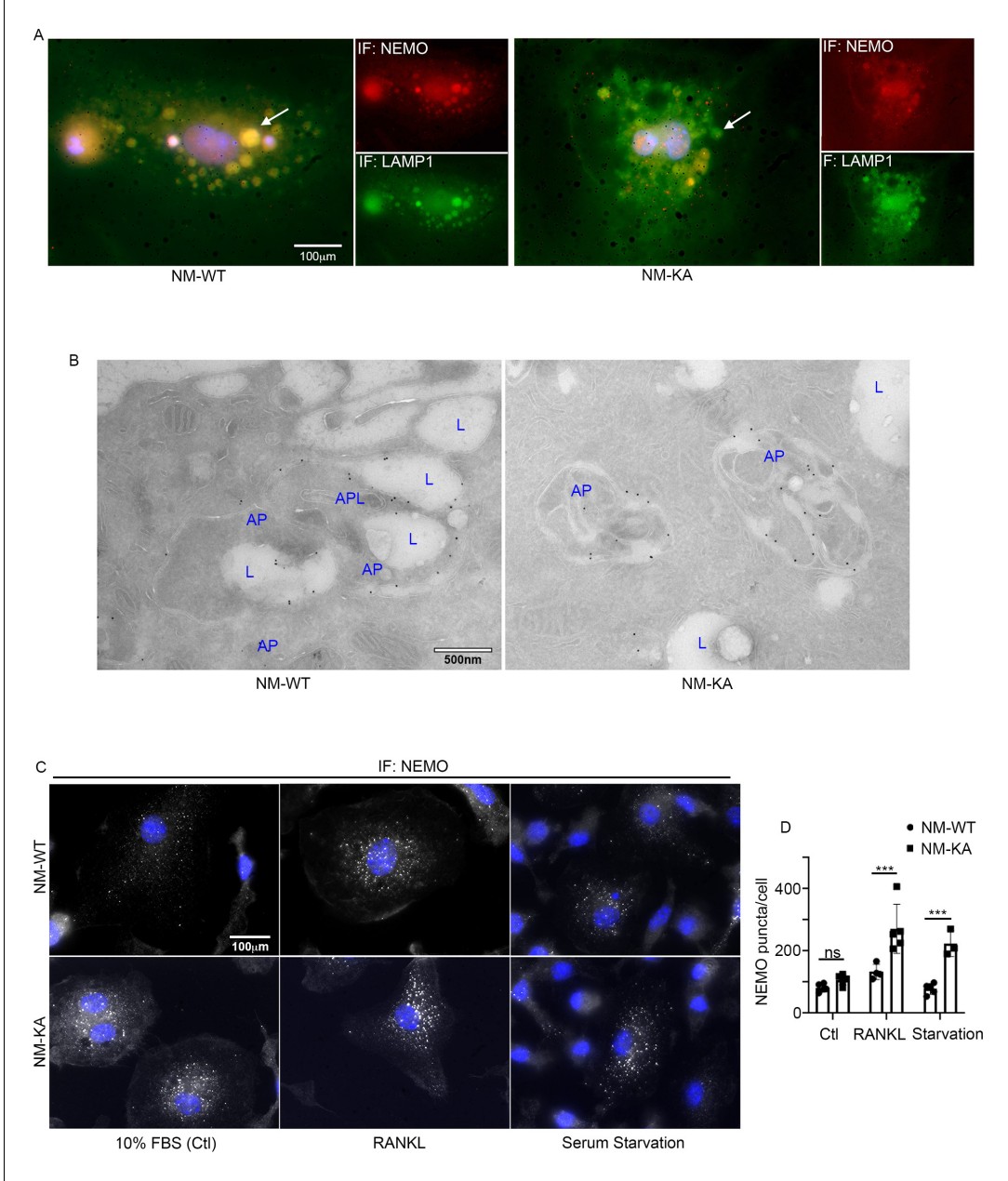

**Figure 5.** NEMO[K270A] is restricted to autophagosomes whereas NEMO[WT] is delivered to lysosomes. preOC from NM-WT and NM-KA mice were pelleted and processed for Immunofluorescence (IF) and electron microscopic analyses after 6 hr of serum starvation. (**A**) Representative IF images showing NEMO (red) and LAMP1 (green). Arrows indicate colocalization of NEMO in LAMP1 positive vacuole-like structures in NM-WT, which is decreased in NM-KA preOC. (**B**) Representative electron microscopic images (x7500) lysosome (L), Autophagosome (AP) and APL (Autophagolysosome). (**C**) Representative IF images showing changes in cellular NEMO organization in response to autophagy induction by serum starvation in NM-WT and NM-KA preOC cells. NEMO-puncta (white) and nucleus (blue). (**D**) NEMO-puncta quantification. *LysM-cre-NEMO-WT-f/f* (NM-WT), *LysM-cre-NEMO-K270A-f/f* (NM-KA) mice. (*p<0.05, **p<0.01 and ***p<0.001).

The online version of this article includes the following source data and figure supplement(s) for figure 5:

**Source data 1.** NEMO-puncta quantification.

**Figure supplement 1.** preOC from NM-WT and NM-KA mice were processed for Immunofluorescence (IF) analysis after 6 hr of serum starvation.

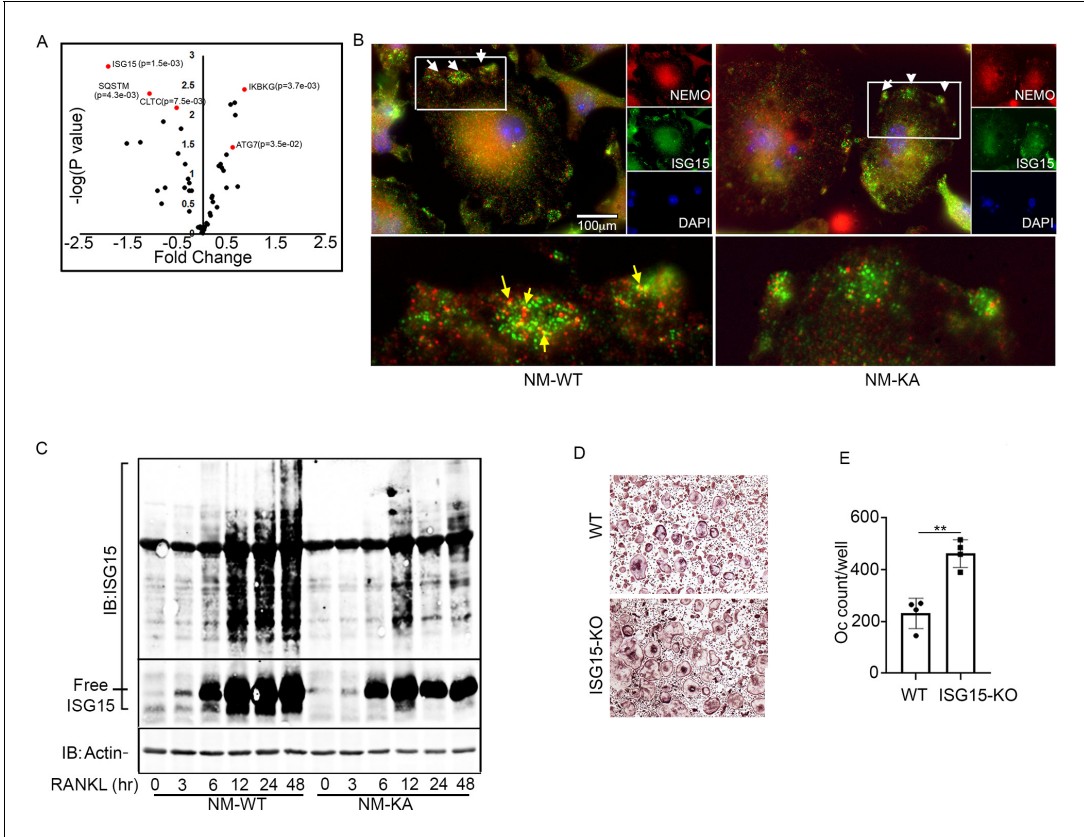

**Figure 6.** Intact NEMO K270 residue is essential for post-translational modification (PTM) by ISG15. (**A**) Volcano plot showing changes in autophagy and PTM related proteins in immunoprecipitated lysates from NM-WT compared with NM-KA BMMs using anti-NEMO antibody. preOC from NM-WT and NM-KA mice were processed for Immunofluorescence (IF) and Immuno-electron microscopy (EM) analysis after 6 hr of serum starvation. (**B**) Representative IF images of NEMO (red) and ISG15 (green) co-localization in preOC. White arrows indicate foci of expression of ISG15 (enlarged inset at bottom of panel B). Yellow arrows indicating NEMO-ISG15 co-localization. (**C**) ISGylated proteins (upper panel) and free ISG15 in response to RANKL treatment. (**D**) BMMs from WT and *ISG15*-KO mice were isolated and cultured in the presence of MCSF (10 ng/ml) and RANKL (50 ng/ml) for four days. Representative TRAP staining for osteoclast (**D**) and quantification (**E**). (**p<0.01).

The online version of this article includes the following source data and figure supplement(s) for figure 6:

**Source data 1.** Proteomic data from immunoprecipitated lysates from NM-WT compared with NM-KA BMMs.

**Source data 2.** Western blots for ISGylated proteins and free ISG15 in response to RANKL treatment.

**Source data 3.** Osteoclast quantification from WT and ISG15-KO in vitro cultures.

**Figure supplement 1.** Representative Immuno-EM images (x7,500) showing localization of NEMO (black arrows) and ISG15 (blue arrow) in NM-WT and NM-KA cells; lysosomes (L), autophagosome (AP).

cells, yet such colocalization was markedly reduced in NEMO^K270A cells (*Figure 6B*). This observation was further confirmed using immunogold EM imaging demonstrating NEMO(18 nm)-ISG15(12 nm) interaction in autophagosome in NEMO^WT cells, which is then delivered into the lysosome structure following cell 6 hr serum starvation (*Figure 6—figure supplement 1*). In contrast, neither NEMO^K270A-ISG15 interaction nor the delivery of NEMO^K270A to lysosomes were detected in NEMO^K270A cells (*Figure 6—figure supplement 1*). Next, we provide biochemical evidence that whereas RANKL induced robust ISGylation in WT cells, this PTM was markedly reduced in NEMO^K270A lysates (*Figure 6C*), affirming the key role of NEMO post translational ISGylation in osteoclastogenesis, a novel finding that has not be described previously. Indeed, supporting its role as regulator of osteoclastogenesis, BMMs derived from ISG15 null mice generated far more osteoclasts than their WT counterparts (*Figure 6D–E*). This observation was further supported by in vivo data showing that bone mass (BV/TV) of mice lacking ISG15 is significantly lower than WT littermates, a finding further confirmed by increased levels of serum levels of TRAPc5b and CTX-1, both markers of bone resorption (data not shown).

## ISGylation of NEMO is essential to restrain osteoclastogenesis

Our findings thus far suggest that ISGylation of NEMO at K270 occurs in response to stimulation of BMMs with RANKL and appears essential for proper autophagy-regulation of NEMO. In addition, exuberant osteoclastogenesis in the absence of ISG15 or in NEMO$^{K270A}$, which is presumably hypo-ISGylated, events that we show lead to defective autophagy, strongly suggest that proper ISGylation of NEMO is required to turn-off NEMO signaling through autophagy to restrain osteoclastogenesis at the opportune time. To offer further support for this paradigm, retroviral expression of ISG15 inhibited osteoclastogenesis in WT cells. In contrast, exuberant osteoclastogenesis by NEMO$^{K270A}$ cells, which we showed irresponsive to ISG15 and exhibit defective autophagy, remained unabated (*Figure 7A–B*). Mechanistically, we conducted flow cytometry analysis of LC3-GFP levels in NEMO$^{WT}$ and NEMO$^{K270A}$ BMMs overexpressing retroviral (pMX)-ISG15 in response to serum starvation-induced autophagy. The data show that LC3I/II levels were significantly reduced in ISG15-infected WT cells compared with minimal reduction in ISG15-infected NEMO$^{K270A}$ cells (*Figure 7C*; compare black dots in top scatters of NM-WT and NM-KA; also compare shift to the left of yellow histogram (arrow) depicting NM-WT compared to negligible shift in orange histograms depicting NM-KA, both overexpressing ISG15). These changes are further quantified in *Figure 7D*, confirming significant reduction of LC3 levels in ISG15 overexpressing WT cells compared with ISG15 overexpressing NEMO$^{K270A}$ cells, an indicative of defective autophagy in the latter cells. These observations suggest that failure of ISG15 to inhibit osteoclastogenesis in NEMO$^{K270A}$ is likely due to its inability to properly tether NEMO$^{K270A}$ to the autophagy-lysosomal machinery. To overcome this predicament, we fused ISG15 to RFP-NEMO$^{K270A}$ and to RFP-NEMO$^{WT}$. Unlike NEMO$^{K270A}$ alone, the ISG15::NEMO$^{K270A}$ fused constructs inhibited osteoclastogenesis (*Figure 7E–F*; fusion construct) and corrected the autophagy flux evident by disappearance of punctate in NEMO$^{K270A}$ cells (*Figure 7G*), reduced LC3 positive puncta and protein (*Figure 7H–I*, *Figure 7—figure supplement 1*), and co-localization of ISG15-NEMO$^{K270A}$ with LAMP1 (lysosomes)(*Figure 7J*). Taken together, forced fusion of ISG15 to NEMO$^{K270A}$ facilitates autophagy and inhibits exuberant osteoclastogenesis.

## Discussion

Previous reports by our group and others have shown that various members of the NF-κB family are essential for osteoclastogenesis and bone homeostasis, whereas their deletion disrupts these processes and leads to skeletal abnormalities (*Abu-Amer and Faccio, 2006*; *Boyce et al., 2010*; *Jimi and Ghosh, 2005*; *Otero et al., 2012*; *Otero et al., 2010*; *Otero et al., 2008*; *Ruocco and Karin, 2005*; *Schett and Smolen, 2005*; *Swarnkar et al., 2016*; *Swarnkar et al., 2014*; *Whyte, 2006*). Generally, the transcription factor NF-κB is activated in response to a multitude of signals in all cell types leading to specific functions that in most cases depend on IKK complex activation. Subsequently, the canonical IKK complex, which is dominated by IKK2 and NEMO activates an array of downstream signals. However, the precise molecular steps underpinning signal to substrate specificity orchestrated by NF-κB in these responses remains unclear. We surmised that the scaffold protein NEMO provides such specificity. Specifically, we deduced that NEMO undergoes signal-specific PTMs at specific residue(s). These PTMs facilitate corresponding biological functions by pairing signal (specifically induced by upstream molecules such as TRAFs, IKKs, etc) with downstream substrates as has been suggested previously (*Schröfelbauer et al., 2012*). This is based on ample reports documenting robust polyubiquitination, SUMOylation, and phosphorylation of NEMO at various lysine and cysteine residues that form selectively in response to different stimuli (*Hay, 2004*; *Liu and Chen, 2011*; *Mabb and Miyamoto, 2007*). More importantly, in most cases, these PTMs mediate destructive or constructive functions such as proteasome-mediated degradation or conversely facilitate intracellular localization and signal transduction (*Fontan et al., 2007*; *Kawadler and Yang, 2006*; *Lamothe et al., 2007*; *Sebban et al., 2006*; *Wu et al., 2006*). In agreement with our hypothesis, and as a proof-of-concept, we uncovered that Lys270 modulate osteoclastogenesis. According to our findings, mutating Lys270 into Ala sensitizes BMMs to RANKL signaling and sustains heightened osteoclastogenesis in vitro and in vivo evident by robust systemic bone loss, skewing toward increased myeloid progenitor frequency and extramodular hematopoiesis (splenomegaly), increased inflammatory burden evident by robust secretion of a myriad of inflammatory mediators, and devastating bone erosion of the joints of mice harboring NEMO$^{K270A}$. Together, these observations suggest that intact K270 in NEMO is critical to restrain and attenuate RANKL

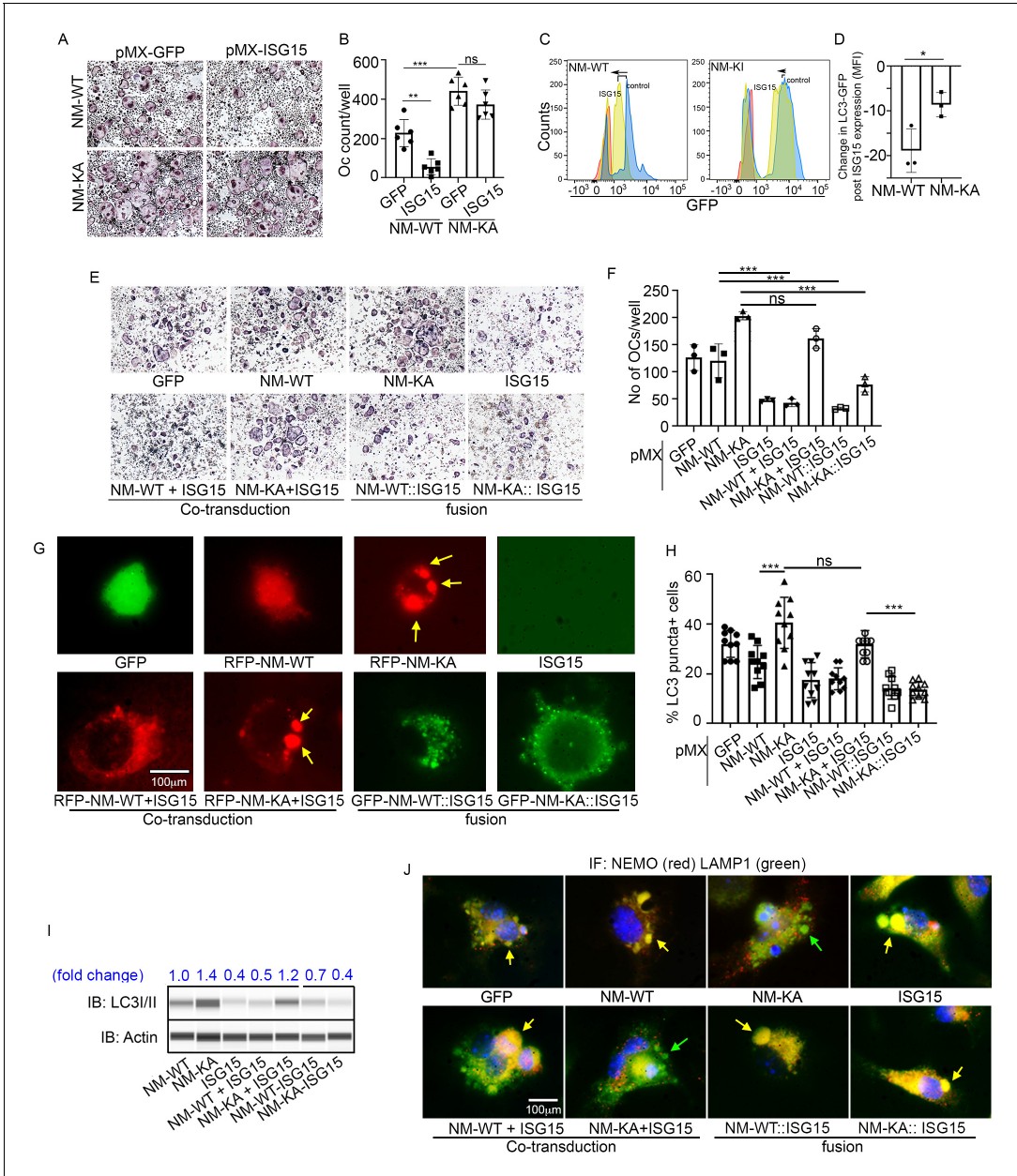

**Figure 7.** ISGylation of NEMO is essential to restrain osteoclastogenesis. BMMs from NM-WT and NM-KA mice were transduced with viral particles (generated by transfecting pMX- retroviral vectors in PLAT-E cells) expressing ISG15 and cultured in the presence of MCSF (10 ng/ml) and RANKL (50 ng/ml) for 4 days. (**A**) Representative TRAP staining for osteoclast (n = 6) and (**B**) quantification of TRAP positive OCs. (**C**) BMMs from NM-WT and NM-KA mice were transduced with ISG15 and *pMRX-GFP-LC3-RFP* retrovirus generated in PLAT-E packing cells. The cells were cultured for 2 days (preOC) followed by 6 hr of serum starvation and flow analysis to detect GFP signal or LC3 flux. (**C**) Histograms representing shift in LC3-GFP+ cells following induction of autophagy. Blue histogram: serum starvation, yellow histogram: serum starvation + ISG15 expression (**D**) Change in Mean Fluorescent Intensity (MFI) showing LC3-GFP signal. (**E**) Wild type BMMs transduced with viral particles (generated by transfecting *pMX*- retroviral vectors in PLAT-E cells) expressing NEMO+/-ISG15, NEMO-K270A+/-ISG15 NEMO-WT::ISG15 (fused) and NEMO-K270A::ISG15 (fused) protein and cultured in the presence of MCSF (10 ng/ml) and RANKL (50 ng/ml) for 4 days.(**E**) Representative TRAP staining for osteoclast (n = 3) and (**F**) quantification of TRAP positive OCs. (**G**) NEMO puncta regulation by ISG15: Live images of preOC expressing RFP-NEMO[WT]+/-ISG15, RFP-NEMO[K270A]+/-ISG15, GFP-NEMO[WT]::ISG15 and GFP-NEMO[K270A]::ISG15 fusion protein. Yellow arrows indicate NEMO[K270A] puncta. ISG15 panel which is not tagged serves as background control. (**H**) Quantification of LC3 puncta+ preOC cells shown in *Figure 7—figure supplement 1*. (**I**) WB for LC3 in preOC expressing NEMO[WT]+/-ISG15, NEMO[K270A]+/-ISG15, NEMO[WT]::ISG15 and NEMO[K270A]::ISG15 fusion protein. (*p<0.05, **p<0.01 and ***p<0.001). (::) denotes fusion. (**J**) Representative IF images (NEMO (Red); LAMP1(green)). NEMO localization in preOC expressing NEMO[WT]+/-ISG15, NEMO[K270A]+/-ISG15, NEMO[WT]::ISG15 and NEMO[K270A]::ISG15 fusion protein. Green arrow- Lysosome and Yellow arrow-localization of NEMO in Lysosome.

The online version of this article includes the following source data and figure supplement(s) for figure 7:

*Figure 7 continued on next page*

*Figure 7 continued*

**Source data 1.** quantification of TRAP positive OCs.
**Source data 2.** LC3-GFP FACS analysis.
**Source data 3.** Quantification of TRAP positive OCs.
**Source data 4.** Quantification of LC3 positive puncta in pre-OC cells shown in *Figure 7—figure supplement 1*.
**Source data 5.** Western blot for LC3 expression in preOC expressing different NEMO and ISG15 constructs.
**Figure supplement 1.** LC3 puncta accumulation of NEMO$^{K270A}$ is reduced by forced fusion with ISG15.

signaling and inflammation. It further suggests that Lys270 serves as a docking site for a RANKL-induced negative feedback mechanism and mutation of this residue renders this regulatory mechanism dysfunctional resulting with robust and unrestrained osteoclastogenesis in vitro and in vivo.

Another significant observation was the apparent enhanced stability and accumulation of NEMO$^{K270A}$ in peri-nuclear cytoplasmic structures. This puncta aggregation of NEMO$^{K270A}$ hinted to us that the protein accumulates and localizes in sub-cellular structures inaccessible for processing reminiscent of failed proteolysis by autophagy. Indeed, using multiple thorough approaches, we show that, unlike NEMO$^{WT}$, NEMO$^{K270A}$ localizes to autophagosomes but failed to co-localize with lysosomes. Consistently, cells expressing this NEMO mutant express high levels of LC3 which accumulated and fails to undergo degradation upon induction of autophagy. In this regard, the link between inflammation, NF-κB, and autophagy has been widely described (*Pawlowska et al., 2018*; *Ravanan et al., 2017*; *Wu and Adamopoulos, 2017*; *Yin et al., 2018*). Several studies have shown that autophagy regulates osteoclast differentiation and joint destruction in experimental rheumatoid arthritis (*Cejka et al., 2010*; *Jaber et al., 2019*; *Lin et al., 2013*; *Sanchez and He, 2009*). Other studies suggested that autophagy is essential to regulate inflammatory responses by reducing levels of inflammatory cytokines (*Wu and Adamopoulos, 2017*), and that dysfunctional autophagy exacerbates skeletal joint disease such as rheumatoid arthritis and osteoarthritis (*Bouderlique et al., 2016*; *Gros, 2017*; *Srinivas et al., 2009*; *Yin et al., 2018*). Hence, autophagy function appears cell context-dependent during physiologic and pathologic conditions.

To decipher the underlying mechanism of this dysfunction, we carried out a proteomic experiment and identified a novel mechanism by which NEMO signal is processed. Indeed, we identified altered expression of major autophagy proteins and some ubiquitin (UB)-like proteins in NEMO$^{K270A}$ cells compared with NEMO$^{WT}$ cells. Most interestingly, we uncovered that levels of the UB-like protein ISG15 were significantly lower in NEMO$^{K270A}$ cells. Given these changes and the similarities between the mechanisms governing the function of ubiquitin, SUMO and ISG15, it was intriguing to identify a potential role for this protein in our system. Unexpectedly, we found that RANKL induces robust expression of ISG15 and ISGylation profile in NEMO$^{WT}$ OCP immunoprecipitates compared with significantly lower ISGylation in NEMO$^{K270A}$ immunoprecipitates.

While the impact of ISGylation on most proteins remains largely unknown, it has been suggested that this modification may regulate proteins by either disrupting or enhancing their activity (*Campbell and Lenschow, 2013*; *Hermann and Bogunovic, 2017*; *Morales et al., 2015*). Despite the scarce knowledge in this field, HEK293 in vitro transfection studies have shown that NF-κB pathway is negatively regulated by ISGylation (*Minakawa et al., 2008*). In other studies, ISG15 was co-localized with the autophagy proteins beclin-1 (BCLN1), HDAC6, and P62/SQSTM1 (*Nakashima et al., 2015*; *Xu et al., 2015*). In one case, ISG15 conjugation of HDAC6 and P62 led to their degradation, providing strong evidence that connects IFN stimulation, ISGylation and autophagy. Moreover, it has been suggested that ISGylation acts as a defense mechanism whereby ISG15 marks proteins to re-direct them towards degradation by the lysosome (*Villarroya-Beltri et al., 2017*). Accordingly, we observed an overall reduction of most autophagy proteins in NEMO$^{K270A}$ immunoprecipitates. Most importantly, we provide direct evidence that forced fusion of ISG15 to hyperactive NEMO$^{K270A}$ facilitates autophagy flux and diminishes osteoclastogenesis. Our findings suggest that ISG15 directly or through other mediators such as ubiquitin chains, tethers target cargo proteins and facilitates fusion with lysosomes leading to their degradation. More specifically, we propose that this process is essential to regulate osteoclastogenesis and attenuate RANKL-induced NF-κB signaling in a timely manner, absence of which leads to unabated osteoclastogenesis and deleterious skeletal anomalies. This is supported by the critical role of NF-κB in general and NEMO specifically in osteoclastogenesis, and by the proposed role of NEMO as a hub not only for

ubiquitin PTMs but also as an interactome with autophagy proteins such as beclin-1, P62, and Rubicon. The significance of ISG15 is further underscored by our observation that BMMs-lacking ISG15 express modestly higher levels of NEMO and LC3, and generate more osteoclasts compared with WT cells. Consistently, ISG15 null mice have moderate osteopenia consistent with overall increased osteoclastogenesis and bone resorption, suggesting that this could be a universal regulatory mechanism. Nevertheless, given the germline deletion of ISG15 in these mice, their overall mild phenotype could be affected by compensatory responses emanating from numerous other cells and tissues which are beyond the scope of this research.

In summary, this study identifies several novel observations. First, we identified NEMO K270 as a crucial regulator of RANKL-induced osteoclastogenesis, and that RANKL appears to utilize this lysine site to exert its osteoclast 'restraining' (negative-feedback) mechanism. Second, we provide novel evidence that mutation of NEMO Lys270 to Ala inflicts an uncontrolled pathologic response that exacerbates osteoclastogenesis. Third, mice harboring myeloid NMEO$^{K270A}$ develop severe osteopenia and joint erosion. Fourth, we identified novel RANKL-induced expression and regulation of ISG15. Consistently, we also provide new evidence that BMMs lacking ISG15 generate more osteoclasts compared with WT littermates, suggesting that ISG15 plays a negative role in this process. Finally, we identified autophagy as key regulatory system recruited by ISG15 through NEMO$^{K270}$ to dampen NF-κB signaling and maintain homeostatic osteoclastogenesis. Altogether, we conclude that ISGylation of NEMO at Lys270 is essential for recruitment of the autophagy machinery to down regulate RANKL signaling.

# Materials and methods

## Key resources table

| Reagent type (species) or resource | Designation | Source or reference | Identifiers | Additional information |
|---|---|---|---|---|
| Strain, Strain background *Mus musculus* | *Ikbkg (Nemo)-floxed* | Dr. Manolis Pasparakis, Cologne, Germany | NM-f/f | C57BL/6 background |
| Strain, Strain background *Mus musculus* | *Ikbkg (Nemo)-K270A-floxed* | Mouse Genetics Core, Washington University in St.Louis | NM-KA-f/f | C57BL/6 background |
| Strain, Strain background *Mus musculus* | *Ikbkg (Nemo)-WT-Tg-floxed* | Mouse Genetics Core, Washington University in St.Louis | NM-WT-Tg f/f | C57BL/6 background |
| Strain, Strain background *Mus musculus* | *Lyz2 (Lysozyme M)-cre* | | LysM-cre | C57BL/6 background |
| Strain, Strain background *Mus musculus* | *LysM-cre-NEMO-flox* | This Paper | NM-cKO | C57BL/6 background |
| Strain, Strain background *Mus musculus* | *LysM-cre-NEMO-K270A-f/f* | This Paper | NM-KA | C57BL/6 background |
| Strain, Strain background *Mus musculus* | *LysM-cre-NEMO-WT-f/f* | This Paper | NM-WT-Tg | C57BL/6 background |
| Strain, Strain background *Mus musculus* | *RELA (NF-KB)-GFP-luciferase reporter* | The Jackson Laboratory | *NF-KB* reporter mice | C57BL/6 background |
| Recombinant DNA reagent | *pMX- retroviral vector* | Cell biolabs | Cat# RTV-010 | Retroviral vector |
| Recombinant DNA reagent | *pMX-GFP* | This paper | | GFP version of pMX retroviral vector |
| Recombinant DNA reagent | *pMX-flag-NEMO-WT-RFP* | This paper | | NEMO WT with flag tag and RFP on pMX backbone-Available in Dr. Yousef Abu-Amer's lab |

*Continued on next page*

*Continued*

| Reagent type (species) or resource | Designation | Source or reference | Identifiers | Additional information |
|---|---|---|---|---|
| Recombinant DNA reagent | *pMX-flag-NEMO-K270A-RFP* | This paper | | NEMO K270A mutant with flag tag and RFP on pMX backbone - Available in Dr. Yousef Abu-Amer's lab |
| Recombinant DNA reagent | *pMX-flag-NEMO-D304N* | This paper | | NEMO D304N mutant on pMX backbone -Available in Dr. Yousef Abu-Amer's lab |
| Recombinant DNA reagent | *pMX-flag-NEMO-K319A* | This paper | | NEMO K319A mutant with Flag tag on pMX backbone -Available in Dr. Yousef Abu-Amer's lab |
| Recombinant DNA reagent | *pMX-flag-NEMO-WT-GFP* | This paper | | NEMO WT with Flag tag and GFP on pMX backbone -Available in Dr. Yousef Abu-Amer's lab |
| Recombinant DNA reagent | *pMX-HA-ISG15* | This paper | | ISG15 with HA tag on pMX backbone -Available in Dr. Yousef Abu-Amer's lab |
| Recombinant DNA reagent | *pMX-flag-NEMO-WT-ISG15-GFP* | This paper | | NEMO WT-ISG15 fusion construct with GFP tag on pMX backbone -Available in Dr. Yousef Abu-Amer's lab |
| Recombinant DNA reagent | *pMX-flag-NEMO-K270A-ISG15-GFP* | This paper | | NEMO K270A-ISG15 fusion construct with GFP tag on pMX backbone -Available in Dr. Yousef Abu-Amer's lab |
| Recombinant DNA reagent | *PMRX-GFP-LC3-RFP retrovirus* | AddGene | Cat# 84573 | LC3 wth GFP and RFP on PMRX backbone |
| Recombinant DNA reagent | Xtreme gene 9 | Roche | Cat# 6365809001 | Transfection reagent |
| Cell line (Homo-sapiens) | PLAT-E | Cell biolabs | Cat# RV-101 | For generating retroviruses |
| Commercial assay or kit | TRAP-Leukocyte kit | Millipore-Sigma | Cat# 387A-1KT | Identify osteoclasts |
| Commercial assay or kit | luciferase activity | GoldBio | Cat# I920-50 | NFkB activity assay |
| Commercial assay or kit | BCA assay | Thermo Fisher | Cat# 23227 | Quantitation of protein |
| Other | Cell lysis buffer | Cell Signaling | Cat# 9803S | Western blot reagent |
| Antibody | donkey anti-rabbit and anti-mouse | LI-COR Biosciences | Cat# 926–32213, RRID:AB_621848 | WB(1:10,000) |
| Antibody | NEMO (Rabbit polyclonal/ Mouse monoclonal) | Santa Cruz | Cat# SC-8330, RRID:AB_2124846 | IF(1:200), WB(1:1000) |
| Antibody | LAMP-1 (Mouse monoclonal) | Santa Cruz | Cat# SC-20011, RRID:AB_626853 | IF(1:200) |
| Antibody | ISG15 (Mouse monoclonal) | Santa Cruz | Cat# SC-166755, RRID:AB_2126308 | IF(1:200), WB(1:1000) |
| Antibody | phos-p65 (Rabbit polyclonal) | Cell Signaling | Cat# 3031, RRID:AB_330559 | WB(1:1000) |
| Antibody | p65 (Rabbit polyclonal) | Cell Signaling Technology, | Cat# 8242, RRID:AB_10859369 | WB(1:1000) |
| Antibody | LC3 (Rabbit polyclonal) | Cell Signaling Technology, | Cat# 3868, RRID:AB_2137707 | IF(1:200), WB(1:1000) |
| Antibody | Flag (Rabbit polyclonal) | Millipore-Sigma | Cat# F1804, RRID:AB_262044 | WB(1:1000) |
| Antibody | β-actin (Mouse monoclonal) | Millipore-Sigma | Cat# A2228, RRID:AB_476697 | WB(1:5000) |

*Continued on next page*

*Continued*

| Reagent type (species) or resource | Designation | Source or reference | Identifiers | Additional information |
|---|---|---|---|---|
| Antibody | anti-B220 (Rat monoclonal) | Thermo Fisher | Cat# 14-0452-82, RRID:AB_467254 | FACS (1 μL per test) |
| Antibody | anti-CD3e (Armenian hamster monoclonal) | Biolegend | Cat# 100301, RRID:AB_312666 | FACS (1 μL per test) |
| Antibody | anti-Gr1 (Rat monoclonal) | Thermo Fisher | Cat# 14-5931-82, RRID:AB_467730 | FACS (1 μL per test) |
| Antibody | anti-Ter119 (Rat monoclonal) | BD Bioscience | Cat#550565, RRID:AB_393756 | FACS (1 μL per test) |
| Antibody | anti-Sca1 PerCP Cy5.5 (Rat monoclonal) | Thermo Fisher | Cat# 122523, RRID:AB_893621 | FACS (1 μL per test) |
| Antibody | anti-c-Kit APC eFluor 780 (Mouse monoclonal) | Thermo Fisher | Cat# 47-1171-82, RRID:AB_1272177 | FACS (1 μL per test) |
| Antibody | anti-CD34 FITC (Mouse monoclonal) | Thermo Fisher | Cat# 343503, RRID:AB_343503 | FACS (1 μL per test) |
| Antibody | CD16/32 eFluor450 (Rat monoclonal) | Thermo Fisher | Cat# 48-0161-82, RRID:AB_1272191 | FACS (1 μL per test) |
| Antibody | colloidal gold conjugated secondary antibodies | Jackson Immuno Research Laboratories | Cat# 715-205-150, RRID:AB_2340822 | Electron microscopy (1:25) |
| Antibody | Alexa Fluor 568 (goat anti-mouse IgG) | Thermo Fisher | Cat# A11031, RRID:AB_144696 | IF (1:2000) |
| Antibody | Alexa Fluor 488 (goat-anti-rabbit IgG) | Thermo Fisher | Cat# A11034, RRID:AB_2576217 | IF (1:2000) |
| Commercial assay or kit | multiplex mouse cytokine kits | R and D Systems | Cat# AYR006 | Inflammation markers |
| Commercial assay or kit | multiplex mouse cytokine kits | Millipore-Sigma | Cat# MCYTMAG-70K-PX32 | Inflammation markers |
| Commercial assay or kit | RatLaps (CTX-1) EIA | Immunodiagnostic Systems | Cat# AC-06F1 | Serum cross-linked telopeptide of type I collagen (CTX-I)-bone resorption marker |
| Commercial assay or kit | Mouse TRAP (TRAcP 5b) kits | Immunodiagnostic Systems | Cat# SB TR-103 | osteoclast marker |
| Commercial assay or kit | PureLink RNA mini kit | Thermo Fisher | Cat# 12183025 | RNA isolation |
| Other | iTaq universal SYBR green super-mix | BioRad | Cat# 1725120 | Real-Time PCR reagent |
| Sequence-based reagent | *TRAP_F* | IDT | PCR primer | CGACCATTGTTAGCCACATACG |
| Sequence-based reagent | *TRAP_R* | IDT | PCR primer | CACATAGCCCACACCGTTCTC |
| Sequence-based reagent | *CTSK_F* | IDT | PCR primer | ATGTGGGTGTTCAAGTTTCTGC |
| Sequence-based reagent | *CTSK_R* | IDT | PCR primer | CCACAAGATTCTGGGGACTC |
| Sequence-based reagent | *MMP9_F* | IDT | PCR primer | ACTGGGCTTAGATCATTCCAGCGT |
| Sequence-based reagent | *MMP9_R* | IDT | PCR primer | ACACCCACATTTGACGTCCAGAGA |
| Sequence-based reagent | *NFATC1_F* | IDT | PCR primer | CCGGGACGCCCATGCAATCTGTTAGT |
| Sequence-based reagent | *NFATC1_R* | IDT | PCR primer | GCGGGTGCCCTGAGAAAGCTACTCTC |
| Software, algorithm | ImageJ | Imagej.nih.gov | | IF image processing, count |

*Continued on next page*

*Continued*

| Reagent type (species) or resource | Designation | Source or reference | Identifiers | Additional information |
| --- | --- | --- | --- | --- |
| Software, algorithm | GraphPad | Graphpad prism-8 software | Statistical Analysis software | Graph preparation, statistical analysis |

## Animals

*Ikbkg (Nemo)-floxed* (NM-f/f) mice on a C57BL/6 background were provided by Dr. Manolis Pasparakis (Cologne, Germany). The *NEMO-K270A-floxed* and *NEMO-WT-Tg-floxed* mice were generated at the Mouse Genetics Core, Washington University (St. Louis, MO). To generate *NEMO-K270A* and *NEMO-WT-floxed* transgenic mice; cDNA encoding *NEMO-K270A* mutation and *NEMO-WT* preceded by a loxP-flanked STOP cassette was cloned into the ubiquitously expressed *Gt(ROSA)26Sor* locus (*Figure 2—figure supplement 1A–B*). In order to conditionally delete NEMO and express NEMO-K270A or NEMO-WT-Tg in myeloid cells, the *NEMO f/f, NEMO-K270A-f/f and NEMO-WT-Tg f/f* mice were crossed with Lyz2 (LysozymeM)-cre mice to generate *LysM-cre-NEMO-flox* (NM-cKO), *LysM-cre-NEMO-K270A-f/f* (NM-KA) and *LysM-cre-NEMO-WT-f/f* (NM-WT-Tg) respectively. *RELA (NF-KB)-GFP*-luciferase reporter mice were purchased from The Jackson Laboratory (Bar Harbor, ME, USA). *ISG15 (ISG15)* knock-out mice were provided by Dr. Deborah Lenschow (Washington University in St. Louis, MO, USA). All the animals were housed at the Washington University School of Medicine barrier facility. All experimental protocols were carried out in accordance with the ethical guidelines approved by the Washington University School of Medicine Institutional Animal Care and Use Committee.

## MicroCT and X-Ray analysis

6–7 weeks old mice were sacrificed and Intact long bones (femur and tibia) from different animals were isolated. The bones were fixed overnight in 10% neutral buffered formalin. After fixation, they were washed with Phosphate Buffer Saline (PBS) and transferred to 70% ethanol (v/v). After fixation, bones were then scanned using Scanco Medical micro-CT systems (Scanco, Wayne, PA, USA) at the core facility at the Musculoskeletal Research Center at Washington University (St. Louis, MO). Briefly, Images were scanned at a resolution of 20 µm, slice increment 20 µm, voltage 55 kV, current 145 µA and exposure time of 200 ms. After scanning, contours were drawn from the growth plate toward trabecular regions of femur. Approximately 150 slices were analyzed. Later contours were drawn and 3D images were constructed. X-ray analysis of whole body and isolated knee joints were performed using Faxitron Ultra Focus 100 on automatic settings and at 3X and 5X magnification, respectively.

## Histology

6–7 weeks old mice were sacrificed and long bones (femur and tibia) from different animals were isolated. The bones were fixed overnight in 10% neutral buffered formalin. After fixation, bones were then decalcified for 2 weeks in decalcification buffer (14% (w/v) EDTA, NH4OH, pH 7.2), dehydrated in graded ethanol (30–70%), cleared through xylene, and embedded in paraffin. Paraffin sections were stained for TRAP to visualize osteoclasts in the bone sections.

## Transfection and retroviral infection

For exogenous expression studies, various constructs (cDNA) were cloned in retroviral pMX- retroviral vector (Cell biolabs, San Diego, CA). For different studies we generated *pMX-GFP, pMX-flag-NEMO-WT, pMX-flag-NEMO-K270A-RFP, pMX-flag-NEMO-D304N, pMX-flag-NEMO-K319A, pMX-flag-NEMO-WT-GFP, pMX-HA-ISG15, pMX-flag-NEMO-WT-ISG15-GFP, pMX-flag-NEMO-K270A-ISG15-GFP.* To generate retroviral production pMX-vectors were first transfected into PLAT-E cells (Cell biolabs, San Diego, CA) using xtreme gene 9 (Roche, San Francisco, CA, USA), followed by collection of virus containing media for next 2 days. This virus containing media with Polybrene (0.8 mg/ml) was used to transduce primary bone marrow cells.

## Cell culture and osteoclastogenesis

Total bone marrow cells were isolated from the long bones (femur and tibia) and cultured in α-MEM media supplemented with 100 units/mL penicillin/streptomycin and 10% FBS (v/v) with 10 ng/mL M-CSF overnight to separate the adherent cells. One day after isolation, the non-adherent cells were collected and used as enriched bone marrow–derived macrophage (BMMs). BMMs were further cultured with M-CSF (20 ng/mL) and RANKL (50 ng/mL) for 4 days followed by fixation and TRAP staining using TRAP-Leukocyte kit (Sigma, St Louis, MO, USA). To investigate changes in autophagy, BMMs were cultured in M-CSF (20 ng/mL) and RANKL (50 ng/mL) for 2 days and used as pre-osteoclast (preOC) for different experiments. To investigate the effect of exogenous expression of different NEMO, NEMO mutants and ISG15, the BMMs after one day of isolation, were transduced with retroviral particles (generated using PLAT-E cells) and osteoclast differentiation was initiated after 2 day of viral transduction.

## RelA-Luc reporter assay

BMMs isolated form *NF-KB-GFP*-luciferase reporter mice were transduced with different pMX-retroviral particles. One day after transduction, the cells were cultured in the presence of M-CSF for two days, followed by RANKL treatment. Post RANKL transfection, cells were lysed and RelA-luciferase activity was measured using luciferase assay system (GoldBio, St. Louis, MO). The luciferase activity was normalized with total protein concentration (BCA assay, Pierce, Invitrogen).

## Western blot analysis

BMMs and/or pre-OC (BMMs treated with RANKL for 2 days) were lysed in cell lysis buffer (Cell Signaling Technology, Danvers, MA, USA) post treatments. Protein concentration was determined using BCA (Pierce, Invitrogen) and equal amounts of protein was loaded onto SDS-PAGE. After transfer, and blocking in 5% BSA for 1 hr at room temperature, membranes were probed with primary antibodies in 5% BSA in PBS-Tween (1% v/v) for overnight and then washed with PBS-Tween (3x) and probed with secondary antibodies from LI-COR (Odyssey Imager; donkey anti-rabbit and anti-mouse) for 1 hr at room temperature. Membranes were then with PBST (3x) and scanned by using LI-COR Odyssey Imager (LI-COR Biosciences, Lincoln, NE, USA). Western blots were also performed (for LC3 and actin) using capillary-based immunoassay using the Wes-Simple Western method with the anti-rabbit detection module (Protein Simple). Protein expression was measured by chemiluminescence. The NEMO and ISG15 antibody were purchased from Santa Cruz, Dallas, TX, USA; phosp-65, p65 and LC3 antibodies were purchased from Cell Signaling Technology, Danvers, MA, USA; Flag and β-actin was purchased from Sigma, St. Louis, MO, USA.

## Flow cytometer analysis

Single cell suspensions from bone marrow were prepared by flushing the marrow out of femur and tibia of mice injected with BrdU (100 µl of 10 mg/mL solution of BrdU in sterile 1X DPBS) 1 days before sacrifice. Following red blood cell lysis, whole bone marrow cells were stained by Zombie UV dye to distinguish live/dead cells. Then bone marrow cells were resuspended in PBS with 2% FBS (FACS buffer), and further stained with biotin-conjugated lineage Ab cocktail (anti-B220, anti-CD3e, anti-Gr1, anti-Ter119). LSK$^+$ (Lin$^-$Sca1$^+$ckit$^-$) cells were stained with Ab cocktail (anti-Sca1 PerCP Cy5.5, anti-c-Kit APC eFluor 780, anti-CD34 FITC, and CD16/32 eFluor450). All FACS antibodies were purchased from either eBioscience, BioLegend (San Diego, CA, USA) or BD Bioscience (San Diego, CA, USA). Following incubation on ice for 45 min, Ab-labeled cells were washed with FACS buffer and subjected to flow cytometric analysis (BD X-20). Data were analyzed with FlowJo software (Tree Star Inc). To measure autophagy flux, flow cytometry analysis of LC3-GFP levels in NEMO WT and NEMO K270A preOC was performed, in response to autophagy induction by serum starvation. preOC were transduced with PMRX-GFP-LC3-RFP retrovirus generated in PLAT-E packing cells. Cells were serum starved for 6 hr and a flow cytometry analysis was done to detect GFP signal. Data were analyzed using FlowJo V10.1 software.

## Multiplex ELISA

Blood from NM-WT and NM-KA mice were collected from submandibular vein and serum was separated using BD-Microtainer tubes. The serum inflammatory cytokine levels were measured using

multiplex mouse cytokine kits (R and D Systems [Minneapolis, MN, USA] and Millipore [San Diego, CA, USA]). Serum cross-linked telopeptide of type I collagen (CTX-I) and TRAP levels were measured using the RatLaps (CTX-1) EIA and Mouse TRAP (TRAcP 5b) kits (Immunodiagnostic Systems, Boldon, UK) using manufacturer's protocol.

## Quantification of mRNA levels by Real-time PCR

BMMs were cultured in presence of M-CSF (20 ng/mL) and RANKL (50 ng/mL) for 3 or 4 days as indicated in the figures. mRNA was isolated using PureLink RNA mini kit (Ambion, Grand Island, NY, USA) and cDNA were prepared using high capacity cDNA reverse transcription kit (Applied Biosystems). Realtime PCR was carried out on BioRad CFX96 real time system using iTaq universal SYBR green super-mix (BioRad, Hercules, CA, USA). mRNA expressions were normalized using β-actin as a housekeeping gene. The following primers were used for qPCR analysis. (*ACP5*) TRAP-F: CGACCA TTGTTAGCCACATACG, TRAP-R: CACATAGCCCACACCGTTCTC, *CTSK (CathepsinK)*-F: ATG TGGGTGTTCAAGTTTCTGC, *CTSK*-R: CCACAAGATTCTGGGGACTC, *MMP9*-F: ACTGGGCTTAGA TCATTCCAGCGT, *MMP9*-R: ACACCCACATTTGACGTCCAGAGA, *NFATC1*-F: CCGGGACGCCCA TGCAATCTGTTAGT, *NFATC1*-R: GCGGGTGCCCTGAGAAAGCTACTCTC.

## Immuno-Electron microscopy

For immunolocalization at the ultrastructural level, preOC from NM-WT and NM-KA mice were fixed in 4% paraformaldehyde/0.05% glutaraldehyde (Polysciences Inc, Warrington, PA) in 100 mM PIPES/ 0.5 mM MgCl2, pH 7.2 for 1 hr at 4°C. Samples were then embedded in 10% gelatin and infiltrated overnight with 2.3M sucrose/20% polyvinyl pyrrolidone in PIPES/MgCl2 at 4°C. Samples were trimmed, frozen in liquid nitrogen, and sectioned with a Leica Ultra cut UCT7 cryo-ultramicrotome (Leica Microsystems Inc, Bannockburn, IL). Ultrathin sections of 50 nm were blocked with 5% FBS/5% NGS for 30 min and subsequently incubated with indicated primary antibodies for 1 hr at room temperature (Note that I tried some of the labeling with primary antibody overnight at 4°C). Following washes in block buffer, sections were incubated by the appropriate colloidal gold conjugated secondary antibodies (Jackson ImmunoResearch Laboratories, Inc, West Grove, PA) for 1 hr. Sections were stained with 0.3% uranyl acetate/2% methyl cellulose and viewed on a JEOL 1200 EX transmission electron microscope (JEOL USA Inc, Peabody, MA) equipped with an AMT eight megapixel digital camera and AMT Image Capture Engine V602 software (Advanced Microscopy Techniques, Woburn, MA). All labeling experiments were conducted in parallel with controls omitting the primary antibody. These controls were consistently negative at the concentration of colloidal gold conjugated secondary antibodies used in these studies.

## Mass-spectroscopy

### Protein identification

MS raw data were converted to peak lists using Proteome Discoverer (version 2.1.0.81, Thermo-Fischer Scientific) with the integration of reporter-ion intensities of TMT 10-plex at a mass tolerance of ±3.15 mDa (*Werner et al., 2014*). MS/MS spectra with charges +2, +three and +four were analyzed using Mascot search engine (*Perkins et al., 1999*) (Matrix Science, London, UK; version 2.6.2). Mascot was set up to search against a SwissProt database of mouse (version June, 2016, 16,838 entries) and common contaminant proteins (cRAP, version 1.0 Jan. 1 st, 2012, 116 entries), assuming the digestion enzyme was trypsin/P with a maximum of 4 missed cleavages allowed. The searches were performed with a fragment ion mass tolerance of 0.02 Da and a parent ion tolerance of 20 ppm. Carbamidomethylation of cysteine was specified in Mascot as a fixed modification. Deamidation of asparagine, formation of pyro-glutamic acid from N-terminal glutamine, acetylation of protein N-terminus, oxidation of methionine, and pyro-carbamidomethylation of N-terminal cysteine were specified as variable modifications. Peptide spectrum matches (PSM) were filtered at 1% false-discovery rate (FDR) by searching against a reversed database and the ascribed peptide identities were accepted. The uniqueness of peptide sequences among the database entries was determined using the principal of parsimony. Protein identities were inferred using a greedy set cover algorithm from Mascot and the identities containing ≥2 Occam's razor peptides were accepted (*Koskinen et al., 2011*).

## Protein relative quantification

The processing, quality assurance and analysis of TMT data were performed with proteoQ (version 1.0.0.0, https://github.com/qzhang503/proteoQ), a tool developed with the tidyverse approach (https://CRAN.R-project.org/package=tidyverse) under the free software environment for statistical computing and graphics, R (https://www.R-project.org/) and RStudio (http://www.rstudio.com/). Briefly, reporter-ion intensities under 10-plex TMT channels were first obtained from Mascot, followed by the removals of PSM entries from shared peptides or with intensity values lower than 1E3. Intensity of PSMs were converted to logarithmic ratios at base two, in relative to the average intensity of reference samples within a 10-plex TMT. Under each TMT channel, Dixon's outlier removals were carried out recursively for peptides with greater than two identifying PSMs. The median of the ratios of PSM that can be assigned to the same peptide was first taken to represent the ratios of the incumbent peptide. The median of the ratios of peptides were then taken to represent the ratios of the incumbent protein. To align protein ratios under different TMT channels, likelihood functions were first estimated for the log-ratios of proteins using finite mixture modelling, assuming two-component Gaussian mixtures (R package: mixtools:: normalmixEM http://www.jstatsoft.org/v32/i06/). The ratio distributions were then aligned in that the maximum likelihood of the log-ratios are centered at zero for each sample. Scaling normalization was performed to standardize the log-ratios of proteins across samples. To discount the influence of outliers from either log-ratios or reporter-ion intensities, the values between the 5th and 95th percentile of log-ratios and 5th and 95th percentile of intensity were used in the calculations of the standard deviations.

## Informatic and statistical analysis

Metric multidimensional scaling (MDS) and Principal component analysis (PCA) of protein log2-ratios was performed with the base R function stats::cmdscale and stats:prcomp, respectively. Heat-map visualization of protein log2-ratios was performed with pheatmap (Raivo Kolde (2019). pheatmap: Pretty Heatmaps. R package version 1.0.12. https://CRAN.R-project.org/package=pheatmap). Linear modelings were performed using the contrast fit approach in limma (*Ritchie et al., 2015*), to assess the statistical significance in protein abundance differences between indicated groups of contrasts. Adjustments of p-values for multiple comparison were performed with Benjamini-Hochberg (BH) correction.

## Immunofluorescence

Post-treatments preOCs were fixed using 4% para-formaldehyde and 0.1% glutaraldehyde for 20 min at room temperature. Post-fixation the cells were washed using PBS (3x) followed by blocking and permeabilization using 0.5% Goat serum and 0.1% saponin (in PBS). Permeabilized cells were later incubated with Primary antibodies (LC3, NEMO, ISG15 and LAMP1 at 1:200 dilution) and Alexa-Fluor secondary antibodies diluted (1:2000) in antibody incubation buffer (1% BSA in 0.1% Saponin in PBS). Fluorescent images were taken at 40X magnification. The images were analyzed using Image-J software.

## Statistical analysis

Statistical analyses were performed by using Student *t* test and Mann Whitney U test. Multiple treatments were analyzed by using one-way ANOVA. For Serum-cytokines analysis outliers were identified using ROUT method. Values are expressed as mean ± SD. *P* values are indicated where applicable. All the statistical analyses were done using GraphPad Prism software. Double-blind analysis was performed to analyze the IF and EM images. Number of experiment repeats, biological replicates and P values are indicted in figure legends.

## Acknowledgements

This work was supported by NIH/NIAMS R01-AR049192, R01-AR054326, R01-AR072623, (to YA), Biomedical grants #86200 and #85160 from Shriners Hospital for Children (YA), P30 AR057235 NIH Core Center for Musculoskeletal Biology and Medicine (to YA), R01-AR075860 and R21-AR077226 (to JS), and R01-AR064755 and R01-AR068972 (to GM).

## Additional information

### Funding

| Funder | Grant reference number | Author |
| --- | --- | --- |
| National Institutes of Health | AR049192 | Yousef Abu-Amer |
| National Institutes of Health | AR054326 | Yousef Abu-Amer |
| National Institutes of Health | AR072623 | Yousef Abu-Amer |
| National Institutes of Health | AR057235 | Yousef Abu-Amer |
| Shriners Hospitals for Children | 86200 | Yousef Abu-Amer |
| Shriners Hospitals for Children | 85160 | Yousef Abu-Amer |
| National Institutes of Health | AR075860 | Jie Shen |
| National Institutes of Health | AR077226 | Jie Shen |
| National Institutes of Health | AR064755 | Gabriel Mbalaviele |
| National Institutes of Health | AR068972 | Gabriel Mbalaviele |

The funders had no role in study design, data collection and interpretation, or the decision to submit the work for publication.

### Author contributions

Naga Suresh Adapala, Data curation, Formal analysis, Investigation, Methodology, Writing - original draft; Gaurav Swarnkar, Manoj Arra, Data curation, Formal analysis, Investigation, Writing - original draft; Jie Shen, Resources, Formal analysis, Funding acquisition, Investigation, Writing - review and editing; Gabriel Mbalaviele, Resources, Formal analysis, Funding acquisition, Validation, Writing - review and editing; Ke Ke, Data curation, Formal analysis; Yousef Abu-Amer, Conceptualization, Resources, Formal analysis, Supervision, Funding acquisition, Project administration, Writing - review and editing

### Author ORCIDs

Yousef Abu-Amer (iD) https://orcid.org/0000-0002-5890-5086

### Ethics

Animal experimentation: All the animals were housed at the Washington University School of Medicine barrier facility. All experimental protocols were carried out in accordance with the ethical guidelines approved by the Washington University School of Medicine Institutional Animal Care and Use Committee (approval protocol #20190002).

### Decision letter and Author response

Decision letter https://doi.org/10.7554/eLife.56095.sa1
Author response https://doi.org/10.7554/eLife.56095.sa2

## Additional files

### Supplementary files

• Transparent reporting form

### Data availability

The following datasets and raw data is be available on Dryad (https://doi.org/10.5061/dryad.tx95x69tn).

The following dataset was generated:

| | Database and |
| --- | --- |

| Author(s) | Year | Dataset title | Dataset URL | Identifier |
|---|---|---|---|---|
| Adapala NS, Swarnkar G, Arra M, Shen J, Mbalaviele G, Ke K, Abu-Amer y | 2020 | Source Dateset: Inflammatory osteolysis is regulated by site-specific ISGylation of the scaffold protein NEMO | https://doi.org/10.5061/dryad.tx95x69tn | Dryad Digital Repository, 10.5061/dryad.tx95x69tn |

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
