## [Decision Letter]

**Acceptance summary:**

In essence, the manuscript was viewed as significant and timely as it honed down at the level of a single amino acid the mechanism of NEMO in the precise regulation of NF-κB signaling as its relates to inflammation-induced bone loss. The connection to the control of autophagy was also seen as valuable, particularly towards the potential design of new therapies for inflammatory osteolysis.

**Decision letter after peer review:**

Thank you for submitting your article "Inflammatory osteolysis is regulated by site-specific ISGylation of the scaffold protein NEMO" for consideration by *eLife*. Your article has been reviewed by three peer reviewers, one of whom is a member of our Board of Reviewing Editors, and the evaluation has been overseen by Clifford Rosen as the Senior Editor. The following individual involved in review of your submission has agreed to reveal their identity: Fayez Safadi (Reviewer #1).

The following points should help you prepare a revised submission.

Summary:

Overall, the reviewers find that the study represents a novel and comprehensive corpus of observations to delineate, in molecular detail, the action of NEMO on bone during inflammation. The study is well done with appropriate controls, and provides a clear conclusion consistent with the data. The implications extend beyond bone into the pathology of inflammation in general terms, and could initiate novel therapeutic approaches. Several issues need consideration to further strengthen the manuscript.

Essential revisions:

1) The inflammatory response includes other immune cells such as T and B cells. Are these cells affected in this NEMO transgenic model?

2) Does use of an autophagy inhibitor, such as chloroquine (CQ), mimic the effect of NEMO K270A on ISGylation and osteoclastogenesis in WT cells?

3) The role of mTOR in autophagy has been widely described. Is mTOR altered in NEMO-K270A cells?

4) Please elaborate on the interaction of NEMO with components of the autophagosome and lysosome.

5) It would be easier to understand if the Results section had more description regarding the purpose of the experiment at the outset. This is especially true when describing measurements of LC3 levels. It would be helpful to point out that increased LC3 levels can suggest defective autophagy since LC3 is normally degraded in physiological autophagy.

---

## [Author Response]

Essential revisions:1) The inflammatory response includes other immune cells such as T and B cells. Are these cells affected in this NEMO transgenic model?

We have conducted flow cytometry of lymphoid cells and observed changes in the frequency of these cells. This change is likely secondary to the ensuing inflammatory response elicited by the innate myeloid response and may require in depth future investigation, albeit not central to the findings of this manuscript.

2) Does use of an autophagy inhibitor, such as chloroquine (CQ), mimic the effect of NEMO K270A on ISGylation and osteoclastogenesis in WT cells?

Indeed, we have included data (Figure 4—figure supplement 1D) in which we show that CQ mimics the effect of NEMO-K270A on osteoclastogenesis.

3) The role of mTOR in autophagy has been widely described. Is mTOR altered in NEMO-K270A cells?

mTOR has been widely described as a regulator of autophagy. We provide date showing that mTOR is elevated in NEMO-K270A cells (Figure 4—figure supplement 1E).

4) Please elaborate on the interaction of NEMO with components of the autophagosome and lysosome.

Completed.

5) It would be easier to understand if the Results section had more description regarding the purpose of the experiment at the outset. This is especially true when describing measurements of LC3 levels. It would be helpful to point out that increased LC3 levels can suggest defective autophagy since LC3 is normally degraded in physiological autophagy.

Completed.